# Batch size selection by stochastic optimal control

**Jim Zhao**
Department of Computer Science
ETH Zürich, Zürich, Switzerland.
`jimzhao@student.ethz.ch`

**Aurelien Lucchi**
Department of Mathematics
Computer Science University of Basel, Basel, Switzerland.

**Frank Proske**
Department of Mathematics
University of Oslo, Oslo, Norway.

**Antonio Orvieto**
Department of Computer Science
ETH Zürich, Zürich, Switzerland.

**Hans Kersting**
Inria, Ecole Normale Supérieure
PSL Research University, Paris, France.

## Abstract

Stochastic gradient descent (SGD) and its variants are widespread in the field of machine learning. Although there is extensive research on the choice of step-size schedules to guarantee convergence of these methods, there is substantially less work examining the influence of the batch size on optimization. The latter is typically kept constant and chosen via experimental validation.
In this work we take a stochastic optimal control perspective to understand the effect of the batch size when optimizing non-convex functions with SGD. Specifically, we define an optimal control problem, which considers the *entire* trajectory of SGD to choose the optimal batch size for a noisy quadratic model. We show that the batch size is inherently coupled with the step size and that for saddles there is a transition-time $t^*$, after which it is beneficial to increase the batch size to reduce the covariance of the stochastic gradients. We verify our results empirically on various convex and non-convex problems.

## 1 Introduction

Stochastic gradient methods are extremely popular in the field of machine learning [4] [3] [10]. Despite their simplicity, they require carefully tuning some hyper-parameters, such as the step size and the batch size. While the choice of step size has been an extensive area of research, including cyclic step sizes [1][9] or adaptive step sizes, such as AdaGrad [7], RMSProp [20], Adam [12], etc., there has been noticeably less work regarding the choice of the batch size. This is somewhat surprising given that batch size tuning has been shown to have significant advantages over tuning the step size [19][8]. A notable exception is a work by Balles et al. [2] that proposes a greedy batch size selection based on maximizing the bound on the expected gain of a single SGD step. Another work by De et al. [6] proposes different increasing batch-size schedules, motivated by approximately constant SNR in gradient approximations, and provides theoretical guarantees for convergence. [11] theoretically analyze the influence of batch size, step size and gradient covariance on the properties of the achieved minima. They show that the ratio of step size to batch size determines a trade-off between the width of the minima, measured by the trace of the Hessian, and the expected terminal loss.
In contrast to prior work that optimizes greedy objectives, we investigate the use of stochastic optimal control in order to study what is the optimal batch size when considering the *entire* trajectory of the stochastic process. To the best of our knowledge, our approach to selecting the batch size is novel. In fact, stochastic optimal control has only rarely been used in the field of optimization, with the exception of [14], but for the problem of step size selection in one dimension. Their extension to the

multi-dimensional case is based on a local diagonal-quadratic assumption. Similarly, we consider the continuous-time representation of SGD and define a continuous control problem that selects the optimal batch size by solving the related Hamilton–Jacobi-Bellman equation (HJBE) analytically, without requiring a diagonal-quadratic assumption.

Furthermore, we verify the validity of our theoretical analysis empirically on various convex and non-convex problems. We show that the derived optimal batch-size schedule only evaluates 19.7 % as many samples compared to switching to the maximal batch size for a 2D saddle point, and only 48.3 % as many samples compared to switching to the maximal batch size for a saddle point in 40 dimensions with two descent directions.

## 2   Continuous-time models for SGD

A common way to analyze SGD is to model it as a stochastic differential equations (SDE), which is a well-established approach in the field of stochastic approximation [13, 14]. In the field of machine learning, this approach was taken in [16] to examine the stationary distributions of a stochastic process and in [11] to determine factors influencing the minima found by SGD. In [17] the authors use continuous-time models of mini-batch SGD and SVRG to derive convergence bounds.

In the case, where the loss function $f : \mathbb{R}^d \to \mathbb{R}$ can be written as the sum of individual functions $f_i$, each corresponding to some data point $i \in [1, \ldots, n]$, that is $\min_\theta \left[ f(\theta) := \frac{1}{n} \sum_{i=1}^n f_i(\theta) \right]$, an update step in mini-batch SGD is of the form

$$\theta_{k+1} = \theta_k - \alpha\eta\nabla f_{\mathcal{B}_k}(\theta_k), \tag{1}$$

where $\alpha\eta$ is the step-size, in which $\eta$ is the maximal allowed step-size, and $\alpha$ is the adjustment factor as was also done in [14], and

$$\nabla f_{\mathcal{B}_k}(\theta) = \frac{1}{m_k} \sum_{i \in \mathcal{B}_k} f_i(\theta), \tag{2}$$

where $|\mathcal{B}_k| = m_k$ for some $m_k \ll n$. Let the empirical covariance of $\nabla f_i(\theta)$ be denoted by $\Sigma(\theta) := \frac{1}{n} \sum_{i=1}^n (\nabla f_i(\theta) - \nabla f(\theta))(\nabla f_i(\theta) - \nabla f(\theta))^T$, then by the assumption above, the covariance of $\nabla f_{\mathcal{B}_k}(\theta)$ is $\text{cov}(\nabla f_{\mathcal{B}_k}(\theta)) = \Sigma(\theta)/m_k$. From this, an SDE of the following form can be derived (see A.1 for details):

$$d\theta_t = -\alpha\nabla f(\theta_t)dt + \alpha\sqrt{\frac{\eta\Sigma(\theta_t)}{m_t}}dB_t, \tag{3}$$

where $dB_t$ is Brownian motion and $m_t$ denotes the time-dependent batch size Now that a continuous-time model of SGD is derived, different tools such as optimal control theory can be applied to analyze the effect of the batch-size when optimizing non-convex functions.

## 3   Optimal control

Now consider a dynamical system with state vector $x_t \in \mathbb{R}^d$ and a control vector $m_t \in \mathbb{R}^l$

$$\frac{dX_t}{dt} = f(X_t, m_t, t), \quad X_0 = x_0, \tag{4}$$

with a given function $f : \mathbb{R}^d \times \mathbb{R}^l \times \mathbb{R} \to \mathbb{R}^d$. [1] The control $m_t$ is limited to the admissible set $\mathcal{M}$ on the fixed time interval $[0, T]$, which is a time-invariant, closed, and convex subset of $\mathbb{R}^l$. The cost functional being considered is

$$J^{m_t}(x, t) := K(\Psi^{m_t}(t \to T, x)) + \int_t^T L(x_\tau, m_\tau, \tau) \, d\tau, \tag{5}$$

with given functions $K : \mathbb{R}^d \to \mathbb{R}$ and $L : \mathbb{R}^d \times \mathbb{R}^l \times [0, T] \to \mathbb{R}$. $\Psi^m(t \to T, x)$ denotes the forward flow map [18] following the system of ODEs in Eq. (4) with some batch-size schedule $m_t$ and the initial condition $X_t = x$ and ending at $X_T =: \Psi^m(t \to T, x)$. The optimization problem

$$m^* = \arg\min_{m:[0,T] \to \mathcal{M}} J^{m_t}(x_0, 0), \quad \Phi(x, t) := \min_{m:[0,T] \to \mathcal{M}} J^{m_t}(x, t) \tag{6}$$

can be solved via the Hamilton-Jacobi-Bellman equation

$$0 = \frac{\partial\Phi(x, t)}{\partial t} + \min_{m \in \mathcal{M}} \left\{ L(x, m, t) + \nabla_x\Phi(x, t)^T f(x, m, t) \right\}$$
$$\Phi(x, T) = K(\Psi^m(t \to T, x)) \tag{7}$$

In the next section we will formulate a continuous control problem that selects the optimal batch-size by optimizing the desired objective function and derive an optimal batch-size schedule.

---

[1] This function $f(\cdot, \cdot, \cdot)$ describes the dynamical system and is not to be confused with the loss function $f(\cdot)$.

# 4 Method

Consider the objective $f(\theta) = \theta^T A \theta$ with $\theta \in \mathbb{R}^d$ and $A \in \mathbb{R}^{d \times d}$. Moreover, we assume for the sake of simplicity that the $f_i$'s are such that $\Sigma(\theta) = \Sigma = \mathrm{diag}(\sigma_1, \ldots \sigma_d)$ is diagonal with constants $\sigma_i \geq 0$, for $i = 1, \ldots, d$. Then the continuous-time representation of SGD is defined by the update

$$d\theta_t = -\alpha A \theta_t dt + \alpha \sqrt{\frac{\eta \Sigma}{m_t}} dB_t. \tag{8}$$

Further, let $A \in \mathbb{R}^{d \times d}$ be symmetric and thus diagonalizable, i.e. $A = V^T \Lambda V$ with orthogonal $V$ and $\Lambda = \mathrm{diag}(\lambda_1, \ldots, \lambda_d)$. (Otherwise we can choose $A' = \frac{1}{2}(A + A^T)$). Note that the $\lambda_i$ can be both positive and negative.

**Theorem 4.1** *The average dimension-decoupled dynamics are given by (see A.2 for a derivation):*

$$\frac{dg_i(t)}{dt} = -2\alpha \lambda_i g_i(t) + \frac{1}{2} \lambda_i \frac{\alpha^2 \eta \sigma_i}{m_t}, \quad for\ i = 1, \ldots, d \tag{9}$$

*with $\sum_{i=1}^d g_i(t) = \mathbb{E}[(\theta_t)^T A \theta_t] = \mathbb{E}[f(\theta_t)]$, where the expectation is over the Brownian motion.*

In a convex setting, a large batch size is generally desired because it corresponds to a smaller variance of the stochastic gradient, which in turn speeds up the convergence. However, larger batch sizes also come at the expense of more expensive computation time. In the following, we therefore optimize the loss value but also add a term that penalizes the size of the batch. Additionally, we introduce a cost-weight-factor $\gamma \in [0, 1]$ to weigh between using small batch sizes and having a low terminal loss.

The continuous control problem with the above dynamics then becomes

$$\min_{m \in [m_{\min}, m_{\max}]} J(\underline{g}, t) = \min_{m \in [m_{\min}, m_{\max}]} \int_t^T (1 - \gamma) m_\tau d\tau + \gamma \cdot \sum_{i=1}^d \Psi_i^m(t \to T, g_i) \tag{10}$$

$$\text{s.t. } J(\underline{g}, T) = \gamma \cdot \sum_{i=1}^d \Psi_i^m(T \to T, g_i) = \gamma \cdot \sum_{i=1}^d g_i, \tag{11}$$

where we use the short-hand notation $J(\underline{g}, t) := J(g_1, \ldots, g_d, t)$, $m_{\min}$ and $m_{\max}$ are the feasible range of batch sizes with $1 \leq m_{\min} < m_{\max} \leq n$, and $\Psi_i^m(t \to T, g_i)$ is the forward flow map of the respective ODE in Eq. (9) starting at $g_i : g_i(t)$ and ending at $\Psi_i^m(t \to T, g_i) := g_i(T)$. This control problem can be solved via the HJB-equation from which we can derive the following batch-size schedule, depending on the eigenvalues $\lambda_i$ of $f(\theta)$ (see A.3 for the derivation)

**Theorem 4.2** *Depending on the sign of the eigenvalues $\lambda_i$, for $i = 1, \ldots, d$ we have the following batch-size schedule:*

$$m_t^* = \begin{cases} \sqrt{\frac{\eta}{2} \frac{\gamma}{(1-\gamma)} \sum_{i=1}^d \sigma_i \lambda_i e^{-2\alpha \lambda_i(T-t)}} \cdot \alpha & \text{if } f(\theta) \text{ is convex, i.e. } \lambda_i \geq 0\ \forall i \\ m_{min} & \text{if } f(\theta) \text{ is concave, i.e. } \lambda_i \leq 0\ \forall i. \end{cases} \tag{12}$$

*With a non-convex objective $f(\theta)$ we can assume w.l.o.g. that the eigenvalues are ordered, such that $\lambda_1 < \ldots < \lambda_p \leq 0 < \lambda_{p+1} < \ldots < \lambda_d$. Then the batch-size schedule is*

$$m_t^* = \begin{cases} m_{min} & \text{if } t \leq t^* \\ \sqrt{\frac{\eta}{2} \frac{\gamma}{(1-\gamma)} \sum_{i=1}^d \sigma_i \lambda_i e^{-2\alpha \lambda_i(T-t)}} \cdot \alpha & \text{if } t > t^* \end{cases} \tag{13}$$

*with $t^*$ such that*

$$\left| \sum_{i=1}^p \sigma_i \lambda_i e^{-2\alpha \lambda_i(T-t^*)} \right| = \left| \sum_{i=p+1}^d \sigma_i \lambda_i e^{-2\alpha \lambda_i(T-t^*)} \right| \tag{14}$$

In the case where $d = 2$ and $\lambda_1 < 0 < \lambda_2$ we can express $t^*$ explicitly as

$$t^* = T - \frac{1}{2\alpha(\lambda_2 - \lambda_1)} \ln \left( \frac{\sigma_2 \lambda_2}{-\sigma_1 \lambda_1} \right). \tag{15}$$

We can see in Eq. (12) and (13) that the step size is proportionally compensated by the batch size (ignoring $\alpha$ in the exponent), which was already observed in [11] and in [2]. Looking at (15), we see that a change in the schedule only occurs, if $|\sigma_1 \lambda_1| < |\sigma_2 \lambda_2|$. That is, moving along the negative eigendirection is less likely than along the positive eigendirection, which makes it more challenging to escape saddles of this type. In the following we will verify our results empirically.

# 5 Experimental results

We validated our results empirically for a saddle point in 2D, which can be found in Fig. 1. We ran another experiment for $d = 40$, which can be found in A.6. The loss function $f(\theta) = \frac{1}{2}\theta^T A\theta$ was chosen with $A = \text{diag}(-0.001, 0.1)$, such that the positive eigenvalue is two orders of magnitude larger than the negative eigenvalue. The experiment was repeated for 1000 runs of each 2000 iterations. Given the parameter choice in (74), the transition time was calculated to be at $t^* = 1886.22$. The batch-sizes were restricted to the interval $[1, 1000]$. The batch-size returned by the optimal schedule was rounded to the closest integer. A more detailed description of the experiment setup can be found in A.5.

We compared the schedule to using just the minimal/maximal batch-size for the entire run and switching to the maximal batch-size at $t^*$, which corresponds to the case if there is no cost for the size of the batch-size. (An analysis on this can be found in A.4).

Additionally, we compared our schedule to the CABS rule proposed in [2] on a convex quadratic function for $d = 40$ dimensions. We can see in Fig. 2b that our proposed schedule only evaluates 10% as many samples as the CABS rule, but reaches approximately the same average terminal loss. Moreover, if we look in Fig. 2a at the average loss value reached when the CABS rule evaluates as many samples as our proposed schedule we can see that it is 77% higher than the terminal loss achieved by our schedule. This indicates that the learning efficiency with respect to per sample evaluation varies over the course of optimization. Of course in order to fully evaluate learning efficiency other aspects such as the number of parameter updates and the cost of the schedule evaluation needs to be taken into consideration as well.

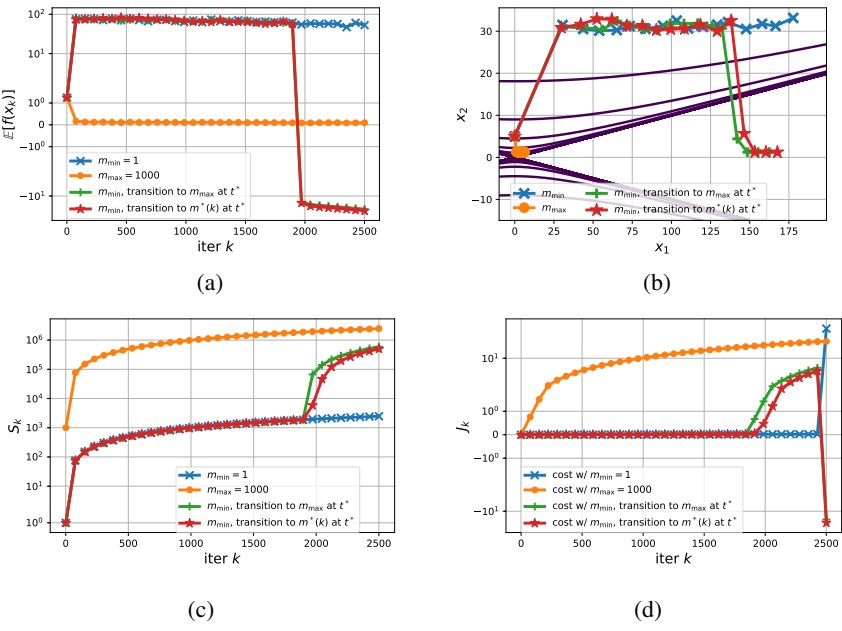

(a)  (b)

(c)  (d)

Figure 1: Averaged quantities over 1000 runs of 2500 iterations each, with the transition time $t^* \approx 1886$. The batch-size schedule was calculated with $T = 2000$, but was kept running for another 500 iterations. (a) Average loss $\mathbb{E}[f(x_k)]$. (b) Contour plot with averaged absolute values of iterates. Every $150^{\text{th}}$ iterate is plotted. Using the maximal batch-size from the beginning (orange) leads to a smooth curve, but does not travel as far along the $x_1$-direction (descent direction). In contrast, switching the schedule (red and green plot) from a small batch size $m_{\min}$ to a larger batch size ($m_{\max}$ or $m^*(k)$) at $t^*$ helps escaping the saddle point. (c) Cumulative number of samples $S_k = \sum_{i=1}^{k} m_i$ evaluated for different schedules. After 2000 iterations only 20.1 % as many samples are evaluated with the adaptive batch-size schedule $m_t^*$ (red) compared to switching directly to $m_{\max}$ (green) at $t^*$. After 2500 iterations the proportion increased to 84.5 %. (d) Average cost $J_k = \mathbb{E}\left[(1-\gamma)\sum_{i=1}^{k} m_i + \gamma \cdot x_{2500}\right]$. Note that both non-constant schedules achieve a negative final loss because the objective function is unbounded from below.

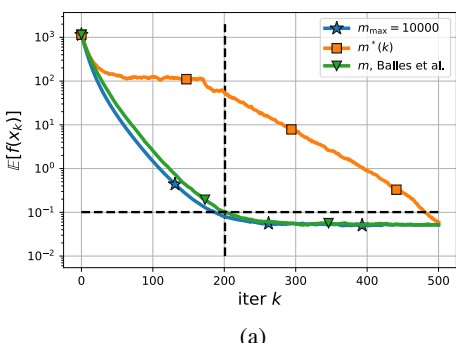
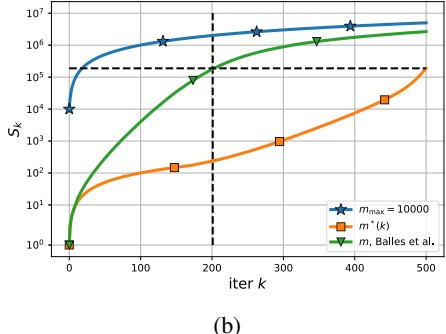

(a)                                 (b)

Figure 2: Comparison of average loss and number of evaluated samples with the CABS rule proposed in [2] over 100 runs of 500 iterations each. [a] Average loss $\mathbb{E}[f(x_k)]$. All three schedules achieve approximately the same terminal loss. The average loss reached by the CABS rule at iteration 201 after evaluating as many samples as the proposed schedule. [b] Cumulative number of samples $S_k = \sum_{i=1}^{k} m_i$ evaluated. At the end of optimization our proposed schedule evalutes only 10% as many sampels as the CABS ruls while reaching almost the same average terminal loss. Moreover, if we compare the average loss value reached when the CABS rule evaluates as many samples as our proposed schedule (after around 257 iterations), we can see that it is still 77% higher than the terminal loss achieved by our schedule.

# 6  Discussion

In this work we tried to understand what the theoretical optimal batch size is, which takes into account the entire trajectory of the continuous-time model of SGD, for optimizing both convex and non-convex noisy quadratic functions in multiple dimensions. In practice, this approach could also be used to optimize more general functions in machine learning, by approximating the function with a local quadratic model. However there are a few limitations to this approach. In practice, the eigenvalues of the Hessian are unknown and need to be estimated. The work of [21] presents a framework in which this could be potentially done. Alternatively, one could optimize quadratic models that are typically used in trast-region methods [5]. Another minor limitation is that the derived schedule is continuous, but a batch-size can only be integer. It is also worth mentioning, that in practice, there are other considerations for choosing the mini batch to achieve optimal performance, for instance to powers of two [15]. Future work could focus on validating the schedule on more general functions and popular machine learning benchmarks, such as MNIST, Cifar-10 or SVHN, where efficient estimation of eigenvalues becomes relevant.

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
