# OpenReview forum: "Batch size selection by stochastic optimal control"
_NeurIPS.cc/2022/Workshop/HITY — HITY Workshop NeurIPS 2022_

### Official Review · Reviewer_NwkE · 2022-10-14
**Important work but practically not useful**

**Rating:** 1
**Confidence:** 2

**Review:**


In this paper, a fundamental problem in training deep learning networks is explored: How to automatically select a good batch size. In contrast to previous studies, the entire trajectory of the stochastic training process is considered here to determine an optimal batch size.


The paper is clearly written and understandable.
The topic might not fit fully to this workshop, since no practically applicable results are given.
The paper is written in a clear and understandable way.
The topic may not be entirely appropriate for this workshop, as no practically applicable results are given.

Unfortunately, I am not familiar with Optimal Control and could not review chapters 3 and 4 in detail.


Points of criticism are:
- Please mention and compare to the approach of [1].
- Please try to find practically useful experiments in addition to your rather simple toy example. Ideally, find a way to estimate the eigenvalues of the Hessian and show how your estimate works on CIFAR-10.


Questions:
- Equation (3): What is the dimension of $\Sigma$? If it is $n \times n$, the dimensions do not match.
- Equation (3): What kind of root operator is this exactly? (element-wise?)
- Figure (1b): It seems to me to be coincidental that the divergence at the beginning leads to better losses. It could also be that the divergence is slightly different and then leads to worse loss as you increase the loss size. Please comment on this.
- Appendix (A) line 187: Your results for the covariance matrix contradict the result of [1]. Please explain why?


Formatting:
- Figure (1) Headings and labels are cut off.



[1] Smith, S. L. and Le, Q. V. (2018). A Bayesian Perspective on Generalization
and Stochastic Gradient Descent. In 6th International Conference on Learning
Representations, ICLR 2018, Vancouver, BC, Canada, April 30 - May 3, 2018,
Conference Track Proceedings. OpenReview.net.

---

### Official Review · Reviewer_3qT2 · 2022-10-19

**Rating:** 1
**Confidence:** 3

**Review:**

Interesting theoretical work, would love to see some simple NN experiments.

---

### Decision · Program_Chairs · 2022-10-20

Accept